



# Results of the third Marine Ice Sheet Model Intercomparison Project (MISMIP+)

Stephen L Cornford[1], Helene Seroussi[2], Xylar S Asay-Davis[3], G Hilmar Gudmundsson[4], Rob Arthern[5], Chris Borstad[6], Julia Christmann[7], Thiago Dias dos Santos[8,9], Johannes Feldmann[10], Daniel Goldberg[11], Matthew J Hoffman[3], Angelika Humbert[7,12], Thomas Kleiner[7], Gunter Leguy[13], William H Lipscomb[13], Nacho Merino[14], Gaël Durand[14], Mathieu Morlighem[8], David Pollard[15], Martin Rückamp[7], C Rosie Williams[4], and Hongju Yu[8]

[1]Centre for Polar Observation and Modelling, Department of Geography, Swansea University, Swansea, UK
[2]Jet Propulsion Laboratory, California Institute of Technology, Pasadena CA, USA
[3]Los Alamos National Laboratory, Los Alamos NM, USA
[4]Faculty of Engineering and Environment, Northumbria University, Newcastle, UK
[5]British Antarctic Survey, Cambridge, UK
[6]Department of Civil Engineering, Montana State University, Bozeman MT, USA
[7]Alfred Wegener Institute for Polar and Marine Research, Bremerhaven, Germany
[8]Department of Earth System Science, University of California, Irvine CA, USA
[9]Centro Polar e Climático, Universidade Federal do Rio Grande do Sul, Porto Alegre RS, Brazil
[10]Potsdam Institute for Climate Impact Research, Potsdam, Germany
[11]Institute of Geography, University of Edinburgh, Edinburgh, UK
[12]University of Bremen, Bremen, Germany
[13] Climate and Global Dynamics Laboratory, National Center for Atmospheric Research, Boulder CO, USA
[14]Université Grenoble Alpes, CNRS, IRD, IGE, Grenoble, France
[15]Earth and Environmental Systems Institute, Pennsylvania State University, USA

**Correspondence:** s.l.cornford@swansea.ac.uk

**Abstract.** We present the result of the third Marine Ice Sheet Intercomparison project, MISMIP+. MISMIP+ is intended to be a test of ice flow models which include fast sliding marine ice streams and floating ice shelves and in particular a treatment of viscous stress that is sufficient for *buttressing*, where upstream ice flow is restrained by a downstream ice shelf. A set of idealized experiments test the models in circumstances where buttressing contributes to a stable steady state, and where a

reduction in that buttressing causes ice stream acceleration, thinning, and grounding line retreat. We find that the most important distinction between models in this particular type of simulation is in the treatment of sliding at the bed, with other distinctions – notably the difference between the simpler and more complete treatments of englacial stress, but also the differences between numerical methods – taking a secondary role.

## 1 Introduction

A number of ice flow models have been developed in the last decade that simulate fast flowing ice streams and ice shelves as well as larger, slower moving ice masses. The key difference between this generation of models and the previous generation is their choice of viscous stress balance equations. All ice sheet models are based upon some approximation to Stokes flow,





but with varying degrees of fidelity (Hindmarsh, 2004; Greve and Blatter, 2009; Pattyn and Durand, 2013). Models designed to simulate the creeping flow of continental ice sheets over glacial cycles are typically based on the shallow ice approxima-

tion (SIA), which considers only vertical shear stresses, while more complete approximations are needed for ice shelves and ice streams. The simplest model that can be applied is the shallow-shelf / shelfy-stream approximation (MacAyeal, 1989), which includes horizontal normal and shear stresses and requires the solution of vertically integrated, two-dimensional stress balance equations. More complete models include the L1Lx class of vertically integrated models (Hindmarsh, 2004; Schoof and Hindmarsh, 2010), which resemble the SSA in many respects, the higher-order (HO) models (Pattyn, 2003) which require

the solution of simplified three-dimensional stress equations, and the complete Stokes models that include all viscous stresses (Le Meur et al., 2004).

There have been several community exercises comparing ice stream and shelf models, and these can be divided into two types: exercises involving real world ice flows, perhaps forced with climate inputs from sophisticated atmosphere and ocean models (Seroussi et al., 2019; Goelzer et al., 2018), and exercises involving idealized settings with simple forcings. This paper

describes the results of an idealized exercise, which can be regarded as sequential to three previous exercises. The Ice Sheet Model Intercomparison for Higher Order Models (ISMIP-HOM, Pattyn et al. (2008)) quantified the differences between SIA, SSA, higher-order, and full Stokes models in time independent settings with periodic bedrock topography and slipperiness. The first Marine Ice Sheet Model Intercomparison Project (MISMIP, Pattyn et al. (2012)) considered a time dependent but laterally unvarying problem and highlighted the technical challenges faced by numerical models of an ice stream with both grounded

and floating portions, that is, with a grounding line. Many models failed to reproduce theoretically well understood properties of such systems. Notably, that a laterally unvarying ice stream on a bedrock that slopes monotonically down in the direction of flow has a single equilibrium state where ice flux across the grounding line matches the total accumulation upstream. The second Marine Ice Sheet Model Intercomparison Project (MISMIP3D, Pattyn et al. (2013)) extended a subset of the MISMIP experiments to include perturbations with some lateral variation, and once again demonstrated that many models of the time did

not exhibit the expected unique equilibrium state. It did not, however, feature strong buttressing of the kind that is important in Antarctic ice shelf and ice stream systems. Recent real world cases include applications to Pine Island Glacier (Joughin et al., 2019) and Thwaites Glacier (Yu et al., 2018) in Western Antarctica, and in Antarctica as a whole (Gudmundsson et al., 2019; Martin et al., 2019).

MISMIP+ explores the ability of ice sheet models to simulate coupled ice sheet and ice shelf systems where the ice shelf

buttresses the flow upstream. All of the experiments are based around an idealized ice stream, adapted from Gudmundsson (2013), with ice sliding into an ice shelf along a bedrock trough with steep walls (Sec. 2.1). Part of the bedrock trough is retrograde – it slopes upward in the direction of flow – and key model parameters are chosen so that, in the absence of sub-ice shelf melting, models should form a stable equilibrium state with the grounding line crossing the center of the channel on this retrograde slope. The equilibrium state is only stable because of lateral variation in the flow field: it is well known that without

such stresses stable steady states form only when the grounding line lies on a prograde slope (Schoof, 2007). The experimental design was stated in (Asay-Davis et al., 2016); in this paper we recap the design for convenience and report the results from fifteen distinct participants, several of whom carried out the experiments with multiple model configuration.





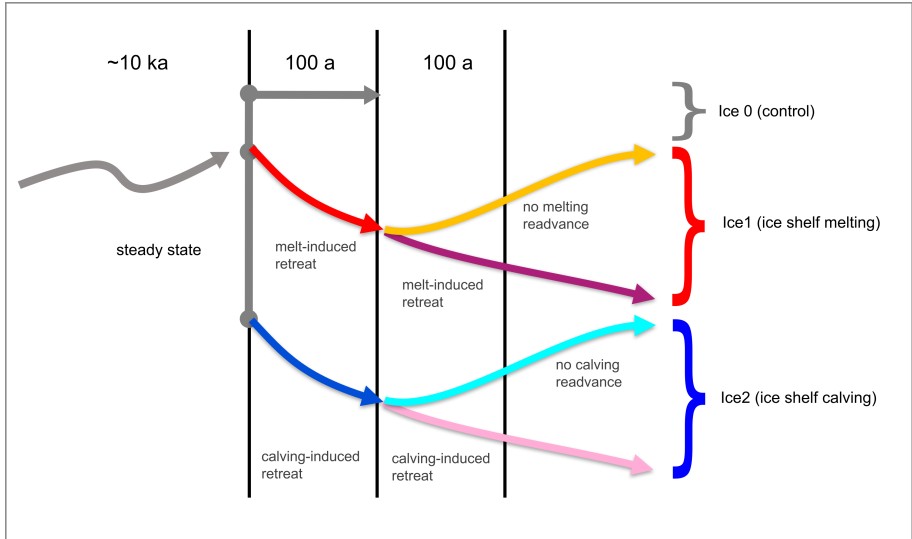

**Figure 1.** Relationship between the MIMSIP+ Ice0, Ice1, and Ice2 experiments. All three experiments start from a common steady state. The Ice 0 experiment provides a control for the Ice1 and Ice2 experiments, which induce retreat with ice shelf melting or calving respectively for 100 a. Later stages of the experiments remove or continue the melting or calving.

## 2 Experimental design

Three groups of experiments (fig. 1) were carried out. The Ice0 experiments (Sec. 2.5) were designed to show that models were close to steady state at the start of the experiments (time $t = 0$). The Ice1 experiments (Sec. 2.6), saw the ice shelves subjected to ablation at the base of the ice, with a simple formula intended to resemble the gross features of melt rates computed by ocean circulation models, with maximum melt rates close to (but not at) the grounding line. Ice shelf ablation also drives the Ice2 experiments, (Sec. 2.7), but in this case the imposed ablation is concentrated at the calving front and does not evolve over time.

### 2.1 Geometry

The MISMIP+ ice stream is set in a rectangular domain, spanning 640 km in the $x$-direction and 80 km in $y$-direction. A no-slip boundary applies at $x = 0$ and free-slip boundaries apply at both lateral boundaries, while calving front boundary conditions apply at $x = 640$ km. This choice of boundary conditions, together with the bedrock geometry and mass sources and sinks, results in solutions that have mirror symmetry about the lateral center of the ice stream. For that reason, some participants chose to run their experiments in only one half of the domain, a perfectly acceptable practice for the MISMIP+ experiments, though not the related ISOMIP+ and MISOMIP experiments (Asay-Davis et al., 2016), where ocean circulation results in non-axisymmetric melt rates. For convenience in this paper, we define $y$ such that $-40 \leq y \leq 40$ km, in contrast with Asay-Davis et al. (2016), so that the axis of symmetry is the $x$−axis.



**Table 1.** Parameter values for the MISMIP+ bedrock geometry

| Parameter | Value |
|-----------|-----------|
| $\bar{x}$ | 300 km |
| $B_0$ | -150 m |
| $B_2$ | -728.8 m |
| $B_4$ | 343.91 m |
| $B_6$ | -50.75 m |
| $w_c$ | 24 km |
| $f_c$ | 4 km |
| $d_c$ | 500 m |

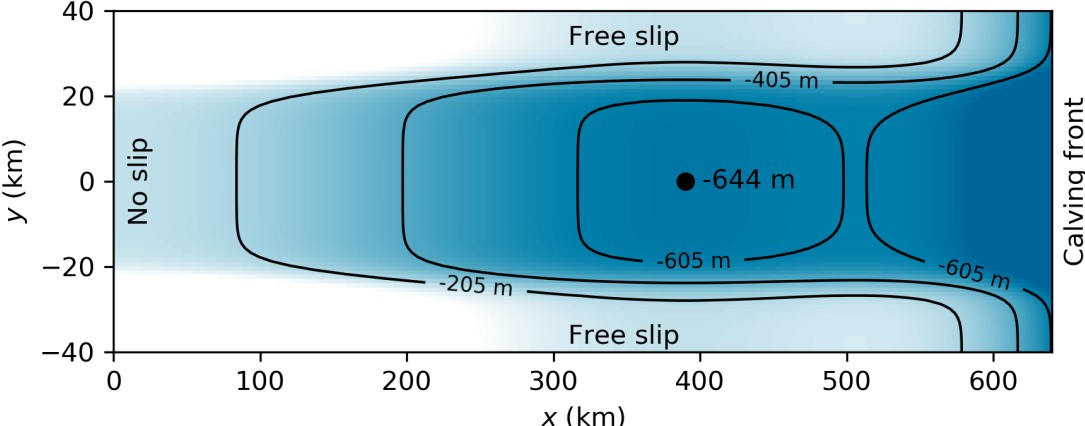

**Figure 2.** MISMIP+ domain showing bedrock elevation $z_b(x,y)$ and boundary types. The spot height (-644 m) indicates the beginning of a retrograde slope in the center of the channel.

The bedrock elevation $z_b$, measured in m, is given by

$$z_b = \max[B_x(x) + B_y(y), -720], \tag{1}$$

where $B_x(x) = -B_0 - B_2 \left(\frac{x}{\bar{x}}\right)^2 - B_4 \left(\frac{x}{\bar{x}}\right)^4 - B_6 \left(\frac{x}{\bar{x}}\right)^6$,

and $B_y(y) = d_c \left[ \left(1 + e^{-2(y-w_c)/f_c}\right)^{-1} + \left(1 + e^{2(y+w_c)/f_c}\right)^{-1} \right]$,

with the parameters value given in Tab. 1. Fig. 2 illustrates the main features of Eq. 1: a steep-walled channel around 48 km wide running parallel to the $x-$axis, and a ridge around $x = 505$ km. Ice flows from the divide at $x = 0$ toward the calving front at $x = 640$ km, so that portion of the ridge between $x = 390$ km and the summit at $x = 505$ km is retrograde: it slopes upward in the direction of ice flow.



## 2.2 Initial state

Participants were asked to compute a steady state given the bedrock geometry and boundary conditions above, a constant rate of accumulation $a = 0.5 \text{ m a}^{-1}$, and no melting at the ice shelf base. No particular method was prescribed. Participants simply needed to show that their initial state was sufficiently close to equilibrium that model drift in experiment Ice0 was small compared to the response to the perturbations of the Ice1 and Ice2 experiments. An obvious if time-consuming method is to carry out a spin-up, evolving the ice sheet from some simple initial state over tens of thousands of model years. Alternatives include taking a steady state from some simpler model and relaxing it over a shorter period of time – a method that might be useful, for example, when working with full Stokes models.

One point of departure from the previous MISMIP designs is the specification of the initial steady state grounding line position, rather than a full set of model parameters. Participants were asked to produce a steady state where the grounding line crossed the centre of the channel ($y = 0$) at a point $x_{gl}(y = 0) = 450 \pm 10$ km; that is, on the retrograde slope. In order to meet this requirement, participants were free to set any value at all for the rate factor $A$, which affects stresses within the ice (Sec. 2.3), and for the basal friction coefficient $\beta^2$, which affects stresses at the ice base (Sec. 2.4). This is a key part of the design for two reasons. First, we wanted all models to begin with their grounding line in steady state on the partly retrograde bed to test that all models could achieve this basic result. Second, we wanted to more closely emulate real world applications of ice sheet models, where the present day geometry of the ice sheet might be well known, but parameters such as $A$ and $\beta^2$ are unknown and are found by some kind of calibration.

## 2.3 Englacial stresses

Participants were permitted to choose any approximation at all for the englacial stresses. That said, the shallow-shelf / shelfy-stream approximation (SSA, MacAyeal (1989)) is the simplest approximation that includes the horizontal normal and shear stresses that describe the coupling between floating and grounded ice, so that in particular we did not expect submissions based solely on the shallow ice approximation (SIA). See, for example, Hindmarsh (2004); Greve and Blatter (2009) for a discussion of stress approximations. Whichever approximation was chosen, it was expected to involve Glen's flow law. In the general case, strain rate components $D_{ij}$ and deviatoric stress components $\tau_{ij}$ satisfy

$$\tau_{ij} = A^{-1/n} D_e^{1/n-1} D_{ij} \tag{2}$$

where $n = 3$ and $2D_e^2 = D_{ij}D_{ji}$. Participants were free to chose any constant $A$ in order to realize a steady state within the specified tolerance, though a suggested value was given: $A = 2.0 \times 10^{-17} \text{ Pa}^{-3} \text{ a}^{-1}$ ($6.34 \times 10^{-25} \text{ Pa}^{-3} \text{ s}^{-1}$).

## 2.4 Basal friction

Participants submitted results from simulations carried out with one or more of three basal friction laws. All three ensure that the basal friction $\boldsymbol{\tau}_b = 0$ for floating ice, but differ in their approach to ice close to flotation, where the effective pressure at the





base, $N$, is low. The simplest of the three is based on the Weertman (1957) rule for sliding over hard beds, with

$$\boldsymbol{\tau}_b = \begin{cases} \beta^2 |\boldsymbol{u}|^{1/3} |\boldsymbol{u}|^{-1} \boldsymbol{u} & N > 0 \\ 0 & N \le 0 \end{cases} \tag{3}$$

Here, $\boldsymbol{u}$ is the horizontal ice velocity at the bed.

Although well known, the Weertman (1957) is certainly not regarded as the final word on glacier sliding (Fowler, 2010; Joughin et al., 2019). Taking a pragmatic (and model-centered) view, it may not be applicable close to the grounding line, where $\boldsymbol{\tau}_b$ will be discontinuous. The two alternative rules ensure that $\boldsymbol{\tau}_b$ is continuous (but not differentiable). Schoof (2005) considered the case of sliding with cavitation, over hard beds, leading to a rule

$$\boldsymbol{\tau}_b = \frac{\alpha^2 \beta^2 N |\boldsymbol{u}|^{1/3}}{\left[ \beta^6 |\boldsymbol{u}| + (\alpha^2 N)^3 \right]^{1/3}} |\boldsymbol{u}|^{-1} \boldsymbol{u} \tag{4}$$

which, among other features, ensures that $|\boldsymbol{\tau}_b|$ does not exceed some fraction of the effective pressure but can be approximated by the Weertman (1957) rule far from the grounding line where $N$ is large. Tsai et al. (2015) considered a case where friction is due to either sliding over hard beds at high $N$ is and deforming beds when $N$ is low. The resulting rule is

$$\boldsymbol{\tau}_b = \min(\alpha^2 N, \beta^2 |\boldsymbol{u}|^{1/3}) |\boldsymbol{u}|^{-1} \boldsymbol{u}. \tag{5}$$

We refer to both modified rules as *Coulomb-limited*, because the Schoof (2005) and Tsai et al. (2015) rules both imply $|\boldsymbol{\tau}_b| \le$
$\alpha^2 N$: in other words, that the magnitude of the basal friction cannot exceed that given by a Coulomb law,

$$\boldsymbol{\tau}_b = \alpha^2 N |\boldsymbol{u}|^{-1} \boldsymbol{u}. \tag{6}$$

Joughin et al. (2019) uses the term *regularized Coulomb*, reflecting another benefit of this class of rules: that they permit Coulomb sliding but do not insist upon it over the whole domain, which would be problematic for two reasons. The first problem is physical: across much of the ice sheet $N \gtrsim 1$ MPa leading to a magnitude of friction larger than observed. The
second problem is mathematical, or at least pragmatic: if $|\boldsymbol{\tau}_b|$ does not increase with $|\boldsymbol{u}|$ anywhere in the domain, then a Dirichlet condition for $\boldsymbol{u}$ must be imposed on at least some part of the horizontal boundary. The modified rules avoid both of these problems by reverting to the familiar Weertman (1957) rule of Eqn. 3 where $N$ is large (or ice flows slowly in the case of Joughin et al. (2019)).

    In all three cases participants were free to modify the parameter $\beta^2$ in order to achieve the desired steady state: the suggested
value was $\beta^2 = 10^4 \, \mathrm{Pa\,m}^{-1/3} \mathrm{a}^{1/3}$ ($3.16 \times 10^6 \, \mathrm{Pa\,m}^{-1/3} \mathrm{s}^{1/3}$).

## 2.5 The Ice0 experiment

The Ice0 experiment is simply a test of the steady state. Simulations ran from $t = 0$ a to $t = 100$ a, with zero sub-ice shelf melt and an upper surface accumulation rate $a = 0.5 \, \mathrm{m\,a}^{-1}$. Since these are the melt and accumulation rates defined for the initial states, models should exhibit little or no variability during this process, or, if they do exhibit some variation, it should result in
little long term drift. That is, fluctuations in ice thickness were acceptable provided that the grounding line did not advance or retreat and the ice volume did not grow or shrink to any great extent compared to the other two experiments.





## 2.6 Oceanic forcing and the Ice1 experiments

The Ice1 experiments were intended to examine the response of models to intense ablation at the base of the ice shelf, with a spatial distribution reflecting the results of typical cavity circulation models (Asay-Davis et al., 2016). The melt rate, measured in $\mathrm{m\,a^{-1}}$, was

$$m_1 = 0.2 \tanh \frac{z_d - z_b}{75} \max(100 - z_d, 0) \tag{7}$$

where $z_d$ is the ice shelf draft and $z_d - z_b$ is the cavity thickness, both measured in metres. The resulting melt rate will generally increase with draft, reaching values of around $100 \mathrm{\ m\,a^{-1}}$ when $z_d \approx 700$ m, but will vanish at the grounding line where the cavity thickness is zero. In the Ice1r experiment, the melt rate was applied to floating ice over the course of 100 a, starting from the steady state at $t = 0$. This results in the loss of much of the original ice shelf thickness over the course of the experiment, and was expected to result in grounding line retreat, assuming that the ice shelf had a role in buttressing the ice upstream. The two follow on experiments, Ice1rr and Ice1ra, were both designed to start from the end of the Ice1 experiment at $t = 100$ a and terminate at $t = 200$ a. Ice1rr is simply a continuation of Ice1r, with the same melt rate applied, while Ice1ra imposes zero melt rate, allowing the ice shelf to thicken, so that the grounding line should re-advance.

## 2.7 Calving and the Ice2 experiments

The Ice2 experiments follow the same basic structure as the Ice1 experiments but impose a different melt rate,

$$m_2 = \begin{cases} 100 \mathrm{\ m\,a^{-1}} & \text{if } x > 480 \text{ km} \\ 0 & \text{otherwise.} \end{cases} \tag{8}$$

This choice of melt rate in Ice2 results in something like a large calving event, where a downstream portion of the ice shelf, amounting to around half the total area, is removed over a short period of time. The majority of the ice removed lies in a zone that provides little buttressing. By allowing a thick ice shelf to form in the wake of the retreating grounding line, the Ice2 experiments test a model's ability to form a new stable steady state with the grounding line on a retrograde slope. They also test the numerical implementation of the model, because there is often an abrupt increase in melt rate immediately downstream from some portions of the grounding line, in contrast to the smooth increase in melt rates seen in the Ice1 experiments.

## 3 Participating models

The participating models cover the same variety of englacial stress approximations as the earlier MISMIP (Pattyn et al., 2012) and MISMIP3d (Pattyn et al., 2013) exercises, with each model including some approximation of the horizontal normal and shear stress (membrane stress). The most complete models make use of the full Stokes equations, but the computational expense entailed by solving the full 3D stress balance equation limits both the number of participants able to run such a model, and the number of submissions from those participants, so that there are only two full Stokes submissions. The most common class of





model is based upon a 2D, vertically-integrated hydrostatic stress balance equation, either through the shallow-shelf / shelfy-stream approximation (SSA), which neglects shear strains $D_{xz}$ and $D_{yz}$ above the bed, or an approximation that assumes a simple form for $D_{xz}$ and $D_{yz}$, for example, the models of Schoof and Hindmarsh (2010) and Goldberg (2011). We will follow (Pattyn and Durand, 2013) in labelling this second class of vertically integrated model 'L1Lx'. Intermediate in complexity are the higher-order (HO) or first-order models, which include the hydrostatic Blatter-Pattyn models. HO models exploit the low

aspect ratio of ice sheets to reduce the four 3D Stokes equations to a pair of 3D equations in the horizontal velocity components. Finally, the HySSA models solve the 2D SSA stress balance equation but adapt that analytic expression (Schoof, 2007) derived for flow-line models with no buttressing to the general case by means of a heuristic buttressing factor: such models were able to produce results in earlier exercises that, in contrast to all other types, did not depend strongly on mesh resolution.

    All of the participating models construct and solve their stress balance and mass transport equations though a limited choice

of methods. There are several finite volume methods and finite difference methods based on rectangular or meshes, extruded vertically and several finite element methods based on unstructured triangular meshes, also extruded vertically. One model (MALI) takes a mixed approach, combining a finite element discretization of the stress balance equation with a finite volume discretization of the mass transport equation. The majority of finite volume and finite difference methods employ spatially uniform meshes, with two exceptions: WAVI employs a wavelet-based adaptive grid to reduce the computational expense in

solving the stress balance equation, while BISICLES makes use of a time-evolving adaptive block structured mesh in both the stress balance and mass transport equations. The finite element methods tend to employ spatially non-uniform meshes that do not change over the course of the simulation, with the exception of two of the several ISSM submissions, which update their meshes over time.

    Models differ in their discretization of the stress balance equations in the region close to the grounding line. One common

class of techniques is the modification of the discretized basal friction term around the grounding line. These techniques have been given at least two different names in the literature *grounding line parameterization* (Gladstone et al., 2010; Leguy et al., 2014) and *sub-element parameterization* (Seroussi et al., 2014b), but they all represent a similar approach. We will use the terms preferred by the individual model authors when referring to their submissions. The simplest type constructs a piece-wise linear approximation to the thickness-above-flotation $(h - h_f)$ and uses that to evaluate a weight, $w \in [0, 1]$, associated with

each grid cell or element that reduces the discrete approximation to the friction term accordingly. None of these schemes introduce additional degrees of freedom, and they also have the same order-of-accuracy with respect to mesh resolution, $O(\Delta x)$, as the unmodified scheme. Nonetheless, they have been seen to improve accuracy for a given mesh resolution in several cases (Feldmann et al., 2014; Seroussi et al., 2014a). A related – but more controversial (Seroussi and Morlighem, 2018) – modification to standard methods applies a similar weighting to the basal melt rate term in the mass balance equation.

The participating models are described briefly below, ordered alphabetically by model name in Sec. 3.1 to Sec 3.11 and listed in Tab. 2. The supplement also contains a data sheet for each model.



**Table 2.** Details of the participating models

| Model (submitter) | Result set | Basal stress | Englacial stress |
|---|---|---|---|
| BISICLES (Cornford) | SCO_BISICLES_L1L2a_Tsai_500m | Tsai | L1Lx |
| | SCO_BISICLES_L1L2b_Tsai_1km | Tsai | L1Lx |
| | SCO_BISICLES_L1L2b_Tsai_250m | Tsai | L1Lx |
| | SCO_BISICLES_L1L2b_Weertman_250m | Weertman | L1Lx |
| | SCO_BISICLES_SSA_Schoof_250m | Schoof | SSA |
| | SCO_BISICLES_SSA_Tsai_250m | Tsai | SSA |
| CISM (Leguy) | GLE_CISM_SSA_Schoof_1km | Schoof | SSA |
| | GLE_CISM_SSA_Weertman_1km | Weertman | SSA |
| Elmer/Ice (Merino) | IME_ElmerIce_FS_Schoof_250m | Schoof | FS |
| | IME_ElmerIce_L1L2b_Schoof_250m | Schoof | L1Lx |
| ISSM (Borstad) | CBO_ISSM_SSA_Tsai_500m | Tsai | SSA |
| ISSM (Seroussi) | HSE_ISSM_HO_Weertman_1km | Weertman | HO |
| | HSE_ISSM_SSA_Tsai_1km | Tsai | SSA |
| | HSE_ISSM_SSA_Tsai_500m | Tsai | SSA |
| | HSE_ISSM_SSA_Weertman_1km | Weertman | SSA |
| ISSM (Yu) | HYU_ISSM_FS_Weertman_500m | Weertman | FS |
| ISSM (Dias dos Santos) | TDI_ISSM_SSA_Tsai_500m | Tsai | SSA |
| | TDI_ISSM_SSA_Weertman_500m | Weertman | SSA |
| ISSM (Christmann) | JCH_ISSM_HO_Tsai_200m | Tsai | HO |
| MALI (Hoffman) | MHO_MPASLI_HO_Weertman_500m | Weertman | HO |
| PISM (Feldmann) | JFE_PISM_SSA+SIA_Tsai_1km | Tsai | L1Lx |
| | JFE_PISM_SSA+SIA_Weertman_1km | Weertman | L1Lx |
| | JFE_PISM_SSA+SIA_Weertman_SG_1km | Weertman | L1Lx |
| | JFE_PISM_SSA+SIA_Weertman_eta_1km | Weertman | L1Lx |
| | JFE_PISM_SSA+SIA_Weertman_eta_SG_1km | Weertman | L1Lx |
| | JFE_PISM_SSA+SIA_eta_Tsai_1km | Tsai | L1Lx |
| | JFE_PISM_SSA_Weertman_SG_1km | Weertman | SSA |
| | JFE_PISM_SSA_Weertman_eta_SG_1km | Weertman | SSA |
| PSU3D (Pollard) | DPO_PSU_HySSA_Weertman_10km | Weertman | HySSA |
| | DPO_PSU_HySSA_Weertman_1km | Weertman | HySSA |
| STREAMICE (Goldberg) | DNG_STREAMICE | Schoof | L1Lx |
| TIMFD3 (Kleiner) | TKL_TIMFD3_HO_Tsai_1km | Tsai | HO |
| Úa (Gudmundsson) | HGU_UA_SSA_Weertman | Weertman | SSA |
| WAVI (Williams) | CWI_WAVI_L1L2c_Weertman_1km | Weertman | L1Lx |
| | CWI_WAVI_L1L2c_Weertman_2km | Weertman | L1Lx |





### 3.1 BISICLES

Six submissions are based on BISICLES (Cornford et al., 2013, 2015), a finite volume model that employs time evolving adaptive mesh refinement to maintain fine resolution close to the grounding line. Three vertically integrated stress approximations

are included: the shallow shelf approximation (SCO_BISICLES_SSA_Schoof_250m, SCO_BISICLES_SSA_Tsai_250m), the Schoof and Hindmarsh (2010) L1L2 approximation (SCO_BISICLES_L1L2a_Tsai_500m), and a modified L1L2 approximation that includes vertical shear in the effective viscosity but neglects it in the mass flux (the remainder). All three basal friction rules are represented. Mesh spacing at the grounding line is set to 250 m in most cases, but two coarser resolution cases are included with $\Delta x \geq 500$ m and $\Delta x \geq 1$ km. All of the submissions apply a one-sided difference when evaluating the gravita-

tional driving stress at the grounding line, with no other parameterization, although a sub-grid friction scheme has been useful in other cases (Cornford et al., 2016).

### 3.2 CISM

Two submissions are based on the CISM (Lipscomb et al., 2013) model. Both employ the shallow-shelf approximation to describe englacial stresses, discretized according to the finite element method, on a uniform mesh with 1 km horizontal res-

olution. Thickness transport is effected with an incremental remappping scheme (Dukowicz and Baumgardner, 2000): ice thickness and velocity data are stores at staggered locations. The difference between submissions is the choice of basal friction rule: one submission uses the Schoof (2005) scheme and the other uses the Weertman (1957) scheme. Both make use of a grounding line parameterization (Leguy et al., 2014).

### 3.3 Elmer/Ice

The two Elmer/Ice (Gagliardini et al., 2013, 2016) submissions differ in their treatment of englacial stresses. IME_ElmerIce_FS_Schoof_250m is a full Stokes model, corresponding to the majority of Elmer/Ice publications, for example Durand et al. (2009); Seddik et al. (2012). IME_ElmerIce_L1L2b_Schoof_250m is a vertically integrated model. Both are finite-element models, and apply a horizontal resolution of 250m (indicated by convergence studies to be adequate) over the region swept out by the grounding line during the Ice1 and Ice2 experiments, and a vertical discretization of 7 layers, with finer

resolution toward the base. Both models employ the modified basal friction law of Eq. 4.

### 3.4 ISSM

Several contributions are based on the ISSM (Ice Sheet System Model), which can treat englacial stresses with a variety of approximations, from the SSA to the full Stokes equations, using the finite element method (Larour et al., 2011). The submissions included include one set of full Stokes simulations (HYU_ISSM_FS_Weertman_500m), treating basal friction with

the Weertman power law model, and two set of Blatter-Pattyn approximation simulations (HSE_ISSM_HO_Weertman_1km, JCH_ISSM_HO_Tsai_200m), treating basal friction with the Weertman power law model and Coulomb-limited basal friction rules respectively, The remaining submissions are all SSA configurations, which use either the Weertman or the Coulomb-





limited basal friction rules, and differ in their treatment around the grounding line. HSE_ISSM_SSA_Tsai_1km, HSE_ISSM_SSA_Tsai_50

and HSE_ISSM_SSA_Weertman_1km all make use of a fixed-in-time, non-uniform mesh of triangular elements that is refined

to either 500 m or 1 km across the region swept out by the grounding line, and employ the SEP1 sub-element parameterization.
CBO_ISSM_SSA_Tsai_500m also relies on fixed, non-uniform mesh, but chooses a different parameterization, SEP2 (Seroussi
et al., 2014b), when evaluating friction in partly grounded elements. Two further submissions, TDI_ISSM_SSA_Tsai_500m
and TDI_ISSM_SSA_Weertman_500m, differ from the others in their use of an evolving adaptive mesh (Santos et al., 2019),
which is updated throughout the simulations to maintain 500 m resolutions close to the grounding line and coarser resolution

elsewhere.

### 3.5   MALI

The MALI (MPAS-Albany Land Ice) (Hoffman et al., 2018) submission treats englacial stresses with the Blatter-Pattyn ap-
proximation and basal friction with the Weertman rule. The stress balance equation is discretized horizonatally in space on an
unstructured mesh of triangular finite elements, while the mass conservation equation is discretized, using the finite volume

method, on the corresponding hexagonal Voronoi tesselation. Time discretization is accomplished with the forward (explicit)
Euler scheme. Basal friction around the grounding line is evaluated by computing it and the flotation criterion at the quadrature
points of a fifth order scheme, leading to a treatment comparable to SEP3 of (Seroussi et al., 2014b). The experiments were
carried out with 500m spatial resolution, while the Ice1r experiment used resolutions from 4 km to 250 m.

### 3.6   PISM

PISM (Bueler and Brown, 2009; PISM authors, 2019; Bueler et al., 2007) employs a uniform-mesh finite volume method
to discretize the mass conservation equation and a finite difference method to discretize the stress balance equation. It uses
either the shallow-shelf approximation (SSA), or the SSA+SIA approximation, which complements SSA sliding with SIA
internal deformation. All eight submissions employ a mesh spacing $\Delta x = 1$ km, and differ in: sub-grid friction interpolation
versus none (Feldmann et al., 2014), SSA versus SSA+SIA, Tsai versus Weertman friction rules, and the use or otherwise

of an ice thickness transformation, $\eta = H^{(2n+2)/n}$ to achieve better numerical results at ice sheet margins (Bueler et al.,
2005). All submission apply a one-sided difference when evaluating the gravitational driving stress at the grounding line. Time
integration is explicit, with the time-step satisfying both an advection CFL criterion determined from the SSA sliding speed
and an additional constraint found by expressing the SIA as a diffusion equation (and so proportional to $\Delta x^2$).

### 3.7   PSU3D

The two PSU3D (Pollard and DeConto, 2012) submissions represent the only HySSA model that took part in MIMSIP+.
This class of models is distinct from the more common types of vertically integrated model in their direct imposition of a
flux across the grounding line, derived from the analytic expression of Schoof (2007), which provides consistent performance
between $\sim 10$ km and $\sim 1$ km. As a result, PSU3D has been able to simulate simulations of Antarctica lasting millions of





years(Pollard and DeConto, 2009). It is a finite-difference model, based on a staggered horizontal grid, and Runge-Kutta time
integration. Note that the ice-cliff failure mechanisms included in current version of this model (Pollard et al., 2015) do not
arise in MISMIP+.

### 3.8   STREAMICE

STREAMICE (Goldberg and Heimbach, 2013) is a physical *package* of the MITgcm climate model (Marshall et al., 1997). It
solves velocities via a finite-element method, using bilinear basis functions on quadrilateral elements on a regular grid; while
its thickness is evolved via an explicit finite volume method. Its vertically integrated stress model is based on Goldberg (2011).
Near the grounding line, basal drag is regularised using a sinusoid profile where thickness is within 5m of flotation (either above
or below) – as this treatment is found to yield reversible grounding line movement in the MISMIP3d experiments. However,
melt is applied to cells **only** where cell-averaged thickness is under flotation.

### 3.9   TIMFD3

TIMFD3 (Kleiner and Humbert, 2014) solves the stress balance equations with the LTSML (Hindmarsh, 2004) higher-order
approximation together with the modified basal friction rule, Eqn. 5, discretized according to the finite difference method. In
comparison to the Blatter-Pattyn approximation, the LTSML approximation considers the vertical resistive stress $R_{zz}$ (van der
Veen and Whillans, 1989) in the momentum balance and thus, vertical longitudinal stresses are not hydrostatic. The MISMIP+
experiments were carried out on a 1 km horizontal grid, with the vertical extent of the ice sheet treated as 9 terrain following
layers, more closely spaced at the base. Coarser resolutions do not exhibit a stable grounding line. The initial state is found by
taking the output from an SSA model (one of the BISICLES submissions with the same selection of $A$ and $\beta^2$) and performing
a 500 year relaxation from that point. The model does not employ any kind of sub-grid interpolation.

### 3.10   Úa

Úa (Gudmundsson et al., 2012) is a finite-element model based on the SSA. It employs an implicit time-stepping method.

### 3.11   WAVI

WAVI (Arthern et al., 2015) is a finite-volume model that makes use of a wavelet-based adaptive grid to accelerate the solution
of the stress balance equation. Its vertically integrated stress model is derived from Goldberg (2011) and treats both membrane
and simplified vertical shear stresses. Sub-grid interpolation of the basal drag, gravitational driving stream, and sub-ice shelf
melt rates are deployed in the finite volumes immediately adjacent to the grounding line. Two complete submissions are
included in this paper, differing only in their uniform mesh resolution. A further two partial submission are included, covering
only the Ice2r experiments. These restrict non-zero sub-ice shelf melt rates to finite volumes whose cell-center is floating.





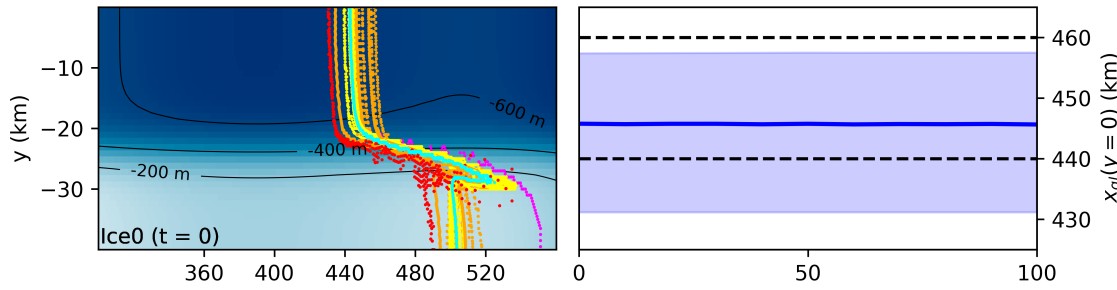

**Figure 3.** Grounding line contours for all models with $\Delta x \leq 1$ km at the start of the Ice0 experiments (left) and variation over 100 years (right). In the left panel, yellow contours correspond to SSA, L1Lx, and HO models with the Coulomb-limited basal friction laws, orange contours to SSA, L1Lx, and HO models with the Weertman basal friction law, red (ISSM) and cyan (Elmer/Ice) contours to full Stokes models, and magenta contours to HySSA models. The color map and black contours depict bedrock elevation. In the right panel, the bold line shows the grounding line position in the center of the channel, $x_{\mathrm{gl}}(y = 0)$, mean averaged over all models, and the shaded region depicts the maximum and minimum values. The dashed lines show the range specified in the experimental design, $x_{\mathrm{gl}}(y = 0) = 450 \pm 10$ km

## 4   Results

The majority of participants completed each experiment, leading to a large volume of results, many of which are in close agreement. The results of each experiment (Ice0, Ice1, and Ice2) are summarised below, concentrating on the spread of results
rather than any individual model. In most cases, a substantial portion of variability in the results can be explained by straight-forward groupings of the participating models: for example models that make use of the (Weertman, 1957) friction rule see slower grounding line migration than models employing either the Schoof (2005) or Tsai et al. (2015) rules. At the same time, distinctions that have been seen to be important in other experiments, such as the use of grounding line parameterization in MISMIP3d (Leguy et al., 2014; Seroussi et al., 2014a; Feldmann et al., 2014), appear unimportant here.

### 4.1   Ice0 experiments

All participating models produced a similar initial grounding line, crossing the channel in a region where the bedrock slopes upward. Fig. 3 plots the grounding line for all models with $\Delta x \leq 1$ km at the start ($t = 0$ a) and the grounding line position $x_{\mathrm{gl}}(y = 0, t)$ in the center of the channel, mean averaged over all models. At the start of the experiments, each of the grounding lines intersects with the channel center-line close to $x_c = 450$ km, and the domain edge close to $x_e = 500$ km, covering most
of the distance between $x_c$ and $x_e$ around the channel walls where the bedrock slopes sharply in the $y-$direction. Most of the models also exhibit a narrow prominence at the top of this lateral bedrock slope, extending downstream from $x_e$. There is little apparent change over time in any model, as required. Not every submission included this test, although the majority did.

Variation between the models is fairly minor. SSA, L1Lx, and HO models that employ the Coulomb-limited basal friction laws are grouped most closely, largely because there was no need to modify the default $A = 2.0 \times 10^{-17}$ Pa$^{-3}$ a$^{-1}$ given in
Asay-Davis et al. (2016). SSA, L1Lx, and HO models that include the Weertman basal friction law are more widely spread,



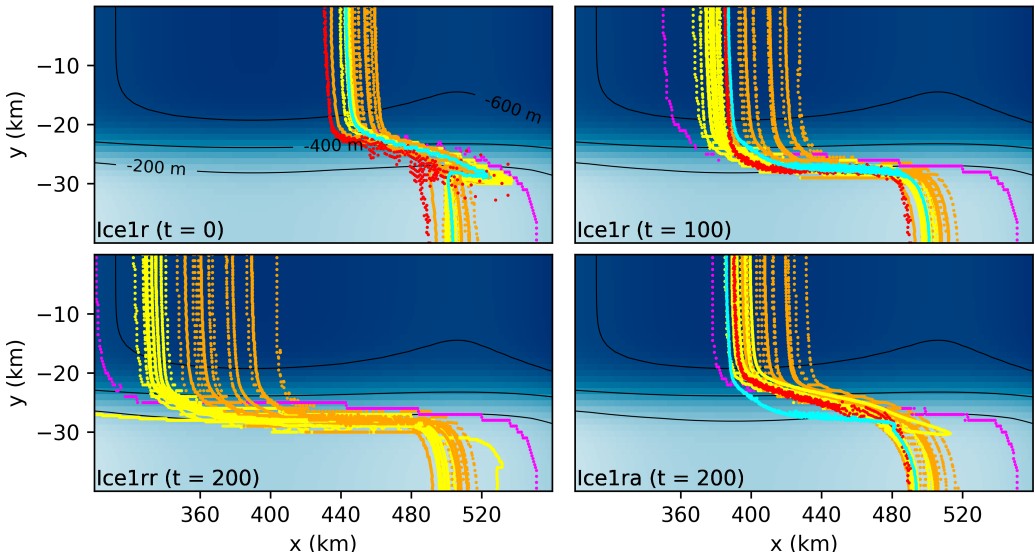

**Figure 4.** Grounding line migration for all models with $\Delta x \leq 1$ km in the Ice1 experiments. The Ice1r experiment starts from the Ice0 steady state position at $t = 0$ a (top left). Grounding lines migrate upstream while the melt rate of Eq, (7) is applied until $t = 100$ a (top right). There are two branches for $t > 100$ a: the Ice1rr branch which continues with the same melt rate (bottom left), and the Ice1ra branch with zero melt rate (bottom right). Yellow contours correspond to SSA, L1Lx, and HO models with the Coulomb-limited basal friction law, orange contours to SSA, L1Lx, and HO models with the Weertman basal friction law, red (ISSM) and cyan (Elmer/Ice) contours to full Stokes models, and magenta contours to HySSA models. The colormap and black contours depict bedrock elevation.

with some models increasing $A$ by as much as 25% to achieve the specified $x_{\mathrm{gl}} = 450 \pm 10$ km, and others submitting results with the grounding line further upstream. Neither of the two full Stokes models elected to modify $A$, but their grounding lines are close to others. Finally, the one HySSA model does produce $x_c \approx 450$ km, setting $A = 3.5 \times 10^{-17}$ Pa$^{-3}$ a$^{-1}$ to do so, but differs from the remaining models over the shallower bedrock with $x_e \approx 540$ km.

## 4.2 Ice1 experiments

Fig. 4 plots the grounding line for all models with $\Delta x \leq 1$ km over the course of the Ice1 experiments. All models see retreat of their grounding line in the centre of the channel while the melt-rate of Eqn. 7 is imposed (the Ice1r and Ice1rr experiments) and little change outside the channel walls beyond the erosion of the prominences. The vast majority of the models also see their grounding lines re-advance once the melt-rate is reduced to zero (the Ice1ra experiment), albeit at a much lower rate so that the initial grounding line is not regained by the end of the experiment at $t = 200$ a. Although all of the models are closely grouped at the start of the Ice1r experiments, a considerable spread of grounding line contours is evident after 100 years. The median retreat of mid-channel grounding line position $x_{gl}(y = 0)$ is in the region of 40 km, but the range is rather larger – close to 100 km. Within groups of models, there is a much smaller spread: less then 10 km between the Coulomb-limited models, a

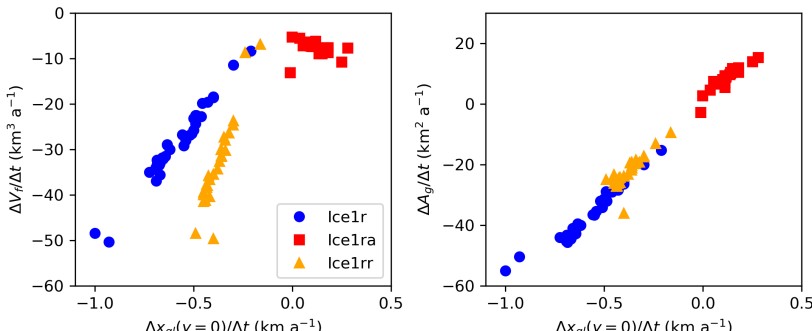

**Figure 5.** Rates of change in mid-channel grounding line position $x_{gl}(y=0)$ plotted against rates of change in volume-above-flotation $V_f$ and grounded area $A_g$. For all three of the Ice1r, Ice1ra, and Ice1rr experiments each point $(\frac{\Delta x_{gl}(y=0)}{\Delta t}, \frac{\Delta V_f}{\Delta t})$ lies close to a straight line, as do the points $(\frac{\Delta x_{gl}(y=0)}{\Delta t}, \frac{\Delta A_g}{\Delta t})$, despite the considerable variation between models.

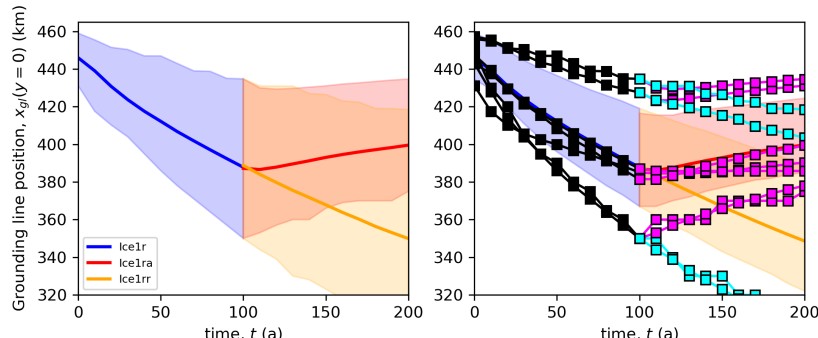

**Figure 6.** Mid-channel grounding line position plotted against time for the Ice1 experiments. The left panel shows the mean (solid curves) and the range (shaded regions) over all models. The right panel shows the mean (solid curves) and range (shaded regions) for the main subset, with individual curves (lines and symbols) for the remainder.

rather larger spread between the Weertman models, and an obvious outlier in the one HySSA model with $\Delta x \leq 1 \, \mathrm{km}$ , whose
mid-channel grounding line position retreats by around 100 km.

We will present the rest of the Ice1 results in terms of the mid-channel grounding line position $x_{gl}(y=0)$, since it represents the other bulk quantities of interest well enough. The rate of change in $x_{gl}(y=0)$ over time represents both the rate of change in volume-above-flotation $V_f$ and the rate of change in grounded area $A_g$ (fig. 5). A single proportional relationship links $x_{gl}(y=0)$ and $A_g$ in the Ice1r, Ice1ra, and Ice1rr experiments. Ice1r and Ice1rr also see proportional relationships between
$x_{gl}(y=0)$ and $V_f$, albeit with distinct constants. The exception is the Ice1ra experiment, where the re-advance of the grounding line following the retreat in Ice1r is associated with a continued, mild loss of volume above flotation.

The majority of models behave similarly in the Ice1 experiments, so we define a number of subsets to represent the spread of results. The first of these is simply the set of all models, but since there are some clear outliers we also define a main

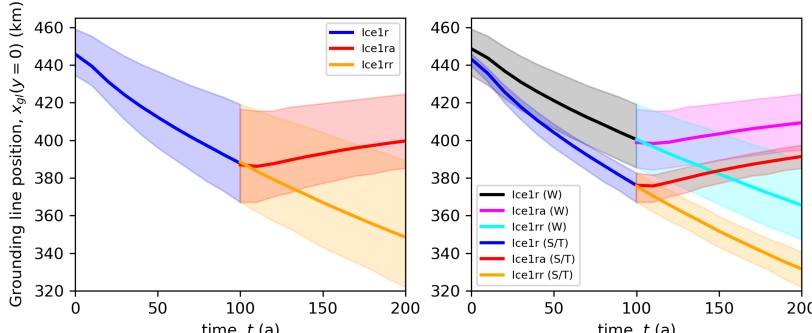

**Figure 7.** Ice1 mid-channel grounding lines position plotted against time for the main subset (left) and the Weertman and Coulomb-limited models (right). Weertman (W) models retreat at 2/3 the rate of Coulomb-limited (S/T) models. The larger spread in the Weertman models is attributed in large part to the spread in initial states

subset comprising models which: (1) completed all experiments, and (2) place their Ice0 steady state grounding line at $x = 450 \pm 15$ km, and (2) see their grounding lines retreat along the centre of the channel to $x = 385 \pm 35$ km. Fig. 6 shows the spread in mid-channel grounding line position against time for all models, the main subset, and the remainder. Of the remainder, two are the HySSA models, and the remaining two employ a distinctive numerical method. We also separate the two full-Stokes models from the lower-order approximations. The main subset then comprises HO, SSA, and L1Lx models with conventional numerical treatments.

One major division within the main subset is the distinction between models that employ the Weertman basal friction law (Eqn. 3) and models the utilise either of the Coulumb limited rules (Eqn. 4) or (Eqn. 5) (fig. 7). The mean rate of retreat seen in the Weertman models is around 0.5 km/a versus 0.7 km/a in the Coulomb-limited models, and indeed the ranges of these two subsets barely overlap. The Weertman subset also exhibits a much larger range, but we attribute this to the larger range of initial states. That larger range is due to only some participants altering the rate factor $A$ to place the initial grounding line within the specified range, rather than any inherent difficulty with implementing the Weertman friction law.

Although the two HySSA simulations achieve an initial state close to the other models, at least in terms of grounding line positions, their transient behaviour differs. Fig. 8 shows both sets of HySSA results with the main subset and the Weertman subset. Their rate of retreat in Ice1r is around 1 km/a, compared to 0.6 km/a for the main subset but it is more appropriate to compare to the 0.5 km/a of the Weertman subset given that both HySSA calculations employ that rule. Something along the same lines was seen in the MISMIP3d experiments (Pattyn et al., 2013). Note though that the HySSA simulations were computed with same model (PSU), albeit at different resolutions (1 km and 10 km), so this may not be a typical result. Note also that the 1 km and 10km HySSA simulations are essentially the same, as has been the case for this class of models in other cases (Pattyn et al., 2013).

The two full-Stokes models see their grounding lines retreat at the same rate as other models while the ice shelf is ablated, but see a much lower rate of advance when the ice shelf regrows. Fig. 9 compares the two full-Stokes models with

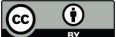



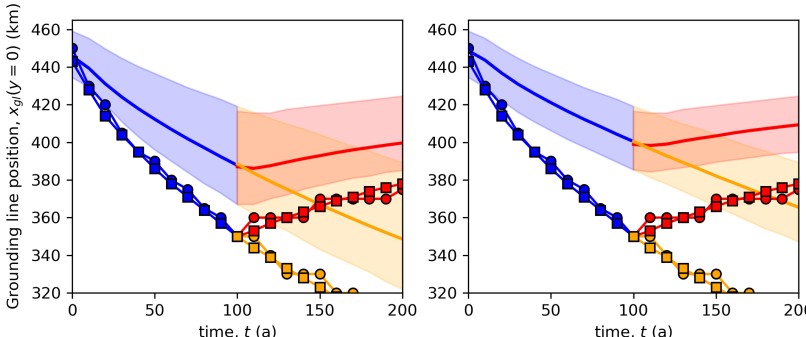

**Figure 8.** Ice1 mid-channel grounding line line position plotted against time for the PSU HySSA models compared to other models. The HySSA models (solid lines with symbols) both exhibit a retreat rate around twice as fast as the main subset mean (shaded regions, left panel), and three times as fast as the Weertman subset mean (shaded regions, right panel).

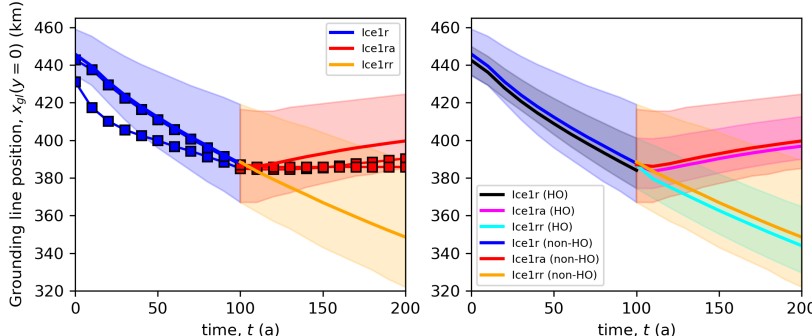

**Figure 9.** Ice1 mid-channel grounding line line position plotted for Stokes, higher-order, and other models. The two full-Stokes models (symbols and lines, left panel) see a comparable rate of retreat to the main subset (shaded regions, left panel) in Ice1r, but a much lower rate of advance in Ice1ra. Within the main subset, higher-order models (HO, right panel) behave in essentially the same way as the other models (non-HO, right panel).

the main subset, and the Ice1r retreat rate of one (IME_ElmerIce_FS_Schoof_250m) lies at the mean of the main sub-set. The other (HYU_ISSM_FS_Weertman_500m) retreats rather more rapidly to begin with, but slows to obtain the same position after 100 years - this may be related to this model's position outside the main subset with regard to its initial state. Neither model sees rapid re-advance when ice shelf ablation ceases in the Ice1ra experiment, though it is notable that

350  IME_ElmerIce_FS_Schoof_250m sees no advance at all. With only two sets of full Stokes results submitted, we are loath to make too much of these differences.

There is little variation between the higher-order and vertically integrated hydrostatic models. Fig. 9 shows that the mean rate of retreat in the ice1r experiment is close to $0.6\,\mathrm{km/a}$ over 100 years for both higher-order (HO) and vertically-integrated (non-HO) models. Likewise, there is essentially no difference between HO and non-HO models in either the mean rate of





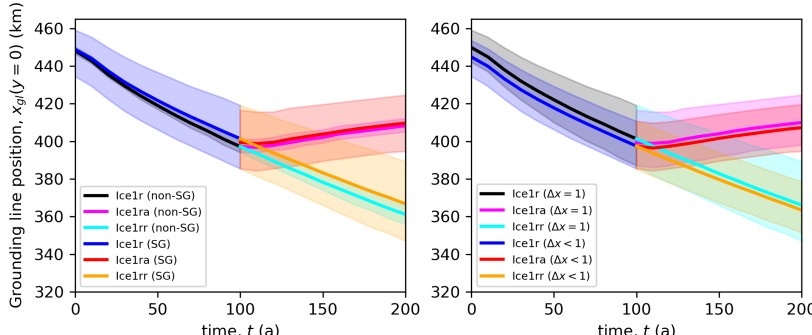

**Figure 10.** Ice1 mid-channel grounding line position plotted for models in the Weertman sub-group with varying numerical treatments. Sub-groups with (SG, left panel) and without (non-SG) sub-grid friction schemes show similar rates of retreat and advance. Likewise, the sub-group with $\Delta x = 1$ km is little different from the sub-group with $\Delta x < 1$ km

further retreat in Ice1rr or the mean rate of re-advance in Ice1ra. There is rather more variability between the non-HO models than between the HO-models, but that can be attributed to the smaller number of HO models.

At least for the $\Delta x \leq 1$km models included in the main subset, the choice between finer resolution or a sub-grid friction scheme is unimportant. Dividing the main subset into sub-groups with and without a sub-grid friction scheme results in the same mean and range of grounding line migration rates, and the same is true if the subset is divided into sub-groups with $\Delta x = 1$ km and $\Delta x < 1$ km (fig. 10). That is not to say that neither a sub-grid scheme nor a fine mesh is consequential in general. Apart from the fact that all models in the main subset have $\Delta x \leq 1$ km, of the 15 models that employ a sub-grid scheme, only 5 have $\Delta x < 1$ km, while of the 11 models that do not employ a sub-grid scheme, 6 have $\Delta x < 1$ km.

### 4.3 Ice2 experiments

The Ice2 experiments are characterized by lower and diminishing rates of retreat compared to the the Ice1 experiments. Fig 11 shows grounding line positions at the start of the Ice2r experiment, before the calving perturbations at $x = 480$ km are applied, 100 years later, and 200 years later both with and without sustained calving at $x = 480$ km. Fig 12 shows the grounding line positions in the center of the channel from $t = 0$ to $t = 100$ a. The average rate of retreat in the first 25 years of the Ice2r experiment is around $0.4 \mathrm{\ km\ a}^{-1}$ but by the last 50 years it has dropped to $0.1 \mathrm{\ km\ a}^{-1}$ as a thick ice shelf is formed downstream of the retreating grounding line. Compare this to the Ice1 experiment, where only a thin ice shelf is formed: the average rate of retreat is initially similar ($0.6 \mathrm{\ km\ a}^{-1}$) but does not decay substantially over time.

The major difference between models in this case is related to numerical methods. The main subset can be split into two subsets: those that exhibit no grounding line retreat along the domain wall, and those that see some retreat. The first group, ice2A, consists solely of models that do not apply a sub-grid interpolation scheme when computing melt rates. The second group, ice2B, includes models that do apply such a scheme, and some that do not but have some treatment special to calving fronts (PISM variants with 'eta' in their name). Fig 13 shows the variation in grounding line position at the domain wall





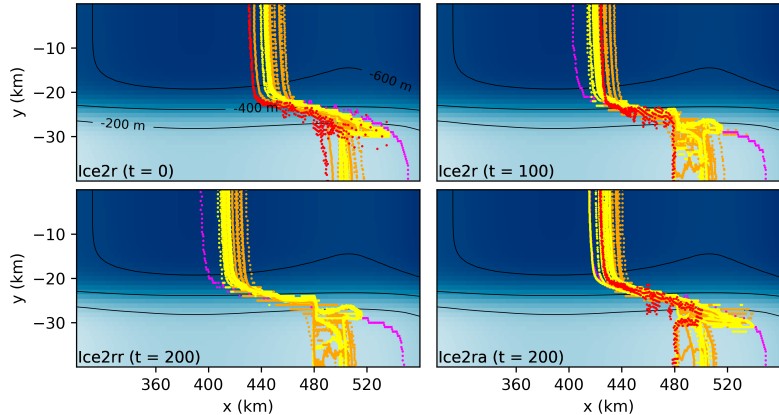

**Figure 11.** Grounding line migration for all models with $\Delta x \leq 1$ km in the Ice2 experiments. The Ice2r experiment starts from the Ice0 steady state position at $t = 0$ a (top left). Grounding lines migrate upstream while the melt rate of Eq, (8) is applied until $t = 100$ a (top right). There are two branches for $t > 100$ a: the Ice2rr branch which continues with the same melt rate (bottom left), and the Ice2ra branch with zero melt rate (bottom right). Yellow contours correspond to SSA, L1Lx, and HO models with the Coulomb-limited basal friction laws, orange contours to SSA, L1Lx, and HO models with the Weertman basal friction laws, red to full Stokes model (ISSM only), and magenta contours to HySSA models. The colormap and black contours depict bedrock elevation

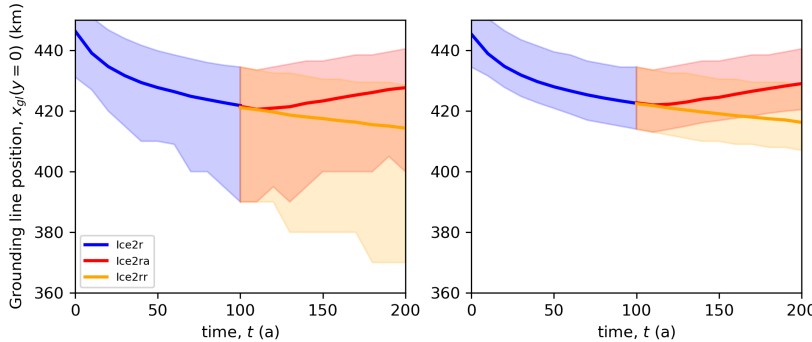

**Figure 12.** Mid-channel grounding line position plotted against time for the Ice2 experiments. The left panel shows the mean (solid curves) and the range (shaded regions) over all models. The right panel shows the mean (solid curves) and range (shaded regions) for the main subset.

between these two groups. Some of the ice2B submissions have their grounding lines retreat to exactly $x = 480$ km – the limit of non-zero melt rate in these experiments – along the wall. All of these do employ a sub-grid interpolation scheme when computing melt rates, and, notably see a retreat rate that grows with their nominal mesh spacing $\Delta x$. For example, the complete WAVI submissions (with sub-grid interpolation of melt rates) exhibit this retreat, whereas the supplementary results (without sub-grid interpolation) do not.

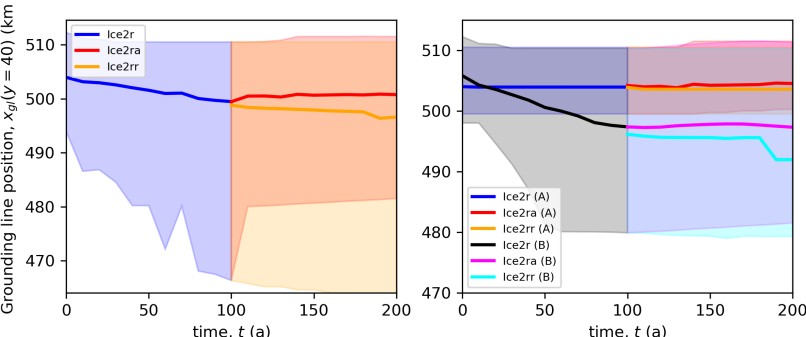

**Figure 13.** Domain edge grounding line position plotted against time for the Ice2 experiments. The left panel shows the mean (solid curves) and the range (shaded regions) for the main subset. The right panel shows the mean (solid curves) and range (shaded regions) for two sub-groups, ice2A and ice2B. Models in sub-group iceA see little or no grounding line retreat along the domain boundary.

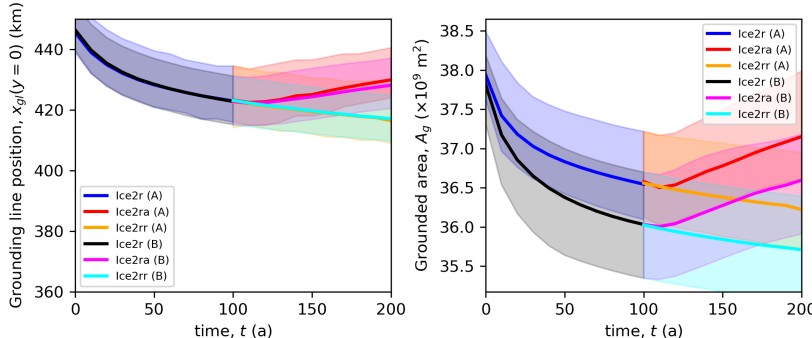

**Figure 14.** Mid-channel grounding line position (left) and grounded area (right) plotted against time for the ice2A and ice2B models. The ice2B models exhibit greater change in grounding line area than ice2a, but that change is restricted to thin ice outside the channel.

The impact of this grounding line retreat along the domain wall, when it does occur, is limited in this case (but is not necessarily limited in others). Fig 14 shows both the grounding line position in the center of the channel and the grounded area as they evolve over the course of the experiment. While the loss of grounded area is significantly greater in the ice2B models, with the mean rate of loss over the ice2B group comparable to the maximum rate in the ice2A group, the vast majority of the additional loss in grounded area is restricted to the thin ice outside of the channel, and there is little difference between the two sub-groups in the center of the channel.

## 5   Discussion

Inasmuch as the MISMIP+ experiments are representative, the choice of basal friction law has more impact on ice stream models than the choice of englacial stress model. Higher order models deliver essentially the same results as vertically integrated treatments, and there is also little variation between the shallow shelf / shelfy stream approximation (SSA) models and





vertically integrated (L1Lx) models that include simplified vertical shear streams. The MISMIP+ experiments are, however, rather biased towards this conclusion, with fast sliding evident across much of the domain. This result appears to extend to the full Stokes models, though we note that only two models of this type were included. On the other hand, models that employ the Coulomb-limited basal friction laws produce faster grounding line retreat and advance compared to models based on the

Weertman rule. A similar conclusion in the context of Thwaites Glacier is reached by Yu et al. (2018)

There is one notable exception to the rule that basal friction physics matter more than englacial physics. The HySSA submissions showed greater rates of grounding line retreat and advance than all of the other model types. This was also observed in the earlier MISMIP3d exercise (Pattyn et al., 2013), but the reasons are not necessarily the same. The HySSA models produced the same steady grounding line position as properly resolved SSA models in MISMIP3d with the same value for $A$, because the

boundary layer approximation they rely upon was derived for that exact case, with no buttressing. In this exercise, the HySSA submissions have $A$ around twice as large, presumably because the buttressing formulation is an informal extension to the boundary layer approximation. It may well be the case that the faster retreat is at least in part due to the slacker grounded ice (with the HySSA models setting the rate factor $A = 3.5 \times 10^{-17}\,\mathrm{Pa^{-3}\,a^{-1}}$ as opposed to the value $A = 2.0 \times 10^{-17}\,\mathrm{Pa^{-3}\,a^{-1}}$ typical of other models) but we note that the result might still carry over to realistic problems because it will be equally

necessary to tune $A$ or its equivalent when matching observations.

Model initial state – in this case the initial ice thickness – is also a key determinant of the response to ice shelf thinning. Models using the Weertman law had a larger spread of initial states than models based on the Coulomb-limited laws, due to model tuning, and they also showed a larger range of retreat rates when subject to strong melt rates. This is within a group of models that all began in similar states, with similar grounding line shapes crossing a retrograde slope, separated by less than

20 km: a greater difference still would be anticipated if any of the models had started from a more obviously distinct state, for example with the grounding line positioned on an entirely pro-grade slope.

Sub-grid treatment of basal friction appears to be of minor importance in these experiments, but does offer considerable benefit in other circumstances. Around half of the participating models employ some sort of modification to the discretization of the basal friction term in the stress balance equation, and around half do not, but there is essentially no difference between

the results of these two groups. This stands in contrast to the MSIMIP3d experiment, where numerical error associated with the abrupt change in basal friction at the grounding line is a major source of differences between models at mesh resolutions comparable to and finer than the $\Delta x \sim 1\,\mathrm{km}$ chosen by most MISMIP+ participants (Pattyn et al., 2013). Convergence studies submitted by participants to MIMSIP+ (see the model datasheets in the supplement) typically indicate that the choice of $\Delta x \sim 1\,\mathrm{km}$ is adequate for MISMIP+, sub-grid friction scheme notwithstanding , while similar studies based around MISMIP3d

reach the opposite conclusion: that a sub-grid friction scheme permits $\Delta x \sim 1\,\mathrm{km}$ and even coarser resolutions while the lack of such a scheme required far finer resolutions (Seroussi et al., 2014a; Feldmann et al., 2014; Gladstone et al., 2010; Leguy et al., 2014). The most obvious difference between the MISMIP3d and MISMIP+ experiments is in the shape of the grounding line and ice shelf, with MISMIP+ having a larger zone of stress transfer between floating and grounded ice, but another possible cause is the colder, stiffer ice of MISMIP3d. MISMIP3d imposed $A \approx 3.0 \times 10^{-18}\,\mathrm{Pa^{-3}\,a^{-1}}$ versus $A \approx 2.0 \times 10^{-17}\,\mathrm{Pa^{-3}\,a^{-1}}$

in MISMIP+, so that shorter length scales in strain-rates are to be expected within the grounding zone.

Sub-grid treatment of melt-rates – as opposed to sub-grid treatment of basal friction – can result in major numerical errors, even at the moderately fine resolutions employed by the majority participating models. The Ice2 experiments show that imposing melt on even a single cell or element that is partly grounded leads to an erroneous melt rate proportional to the mesh spacing $\Delta x$, which is not small compared to the errors caused by under-resolution of the ice dynamics. In other words, if the

mesh is fine enough to obviate the error caused by sub-grid melt schemes, it is also fine enough to to resolve the dynamics. On the basis of these experiments, then, such treatments should be avoided. They are not to be confused with attempts to treat, for example, tidal variation in the grounding line, which may well result in strong melt rates upstream from the mean grounding line but should not produce a retreat rate that vanishes as $\Delta x \to 0$.

## 6   Summary

We have presented the results of the third Marine Ice Sheet Model Intercomparison Project, MISMIP+. It is distinct from its predecessors in that its initial state includes a floating ice shelf that *buttresses* upstream ice, so that when the ice shelf is thinned the grounded ice upstream accelerates and the grounding line retreats, and the converse when the ice shelf is allowed to re-grow. We found that that the most important distinctions between models were the basal friction model and the initial state. In contrast, the distinction between models of varying fidelity, and even between approximate models and full Stokes

models, was minor, at least with the limited range of model resolutions tested. One important exception was the HySSA model, which makes use of an explicit expression for the flux across the grounding line to permit coarse ($\sim 10 \, \text{km}$) resolution and thus simulation over longer time intervals, which exhibited substantially faster grounding line migration than any other model.

*Data availability.* All data used in the paper are included in the supplement at http://doi.org/10.5281/zenodo.3611936

*Competing interests.* No competing interests are present

*Acknowledgements.* The work of Thomas Kleiner has been conducted in the framework of the PalMod project (FKZ: 01LP1511B), supported by the German Federal Ministry of Education and Research (BMBF) as Research for Sustainability initiative (FONA).Support for Matthew Hoffman and the MALI model was provided through the Scientific Discovery through Advanced Computing (SciDAC) program and the Energy Exascale Earth SystemModel (E3SM) project funded by the U.S. Department of Energy (DOE), Office of Science, Biological and Environmental Research, and Advanced Scientific Computing Research programs. This research used resources of the National

Energy Research Scientific Computing Center, a DOE Office of Science user facility supported by the Office of Science of the U.S. Department of Energy under Contract DE-AC02-05CH11231, and resources provided by the Los Alamos National Laboratory Institutional Computing Program, which is supported by the U.S. Department of Energy National Nuclear Security Administration under Contract DE-AC52-06NA25396.



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
