# Peer review of "Results of the third Marine Ice Sheet Model Intercomparison Project (MISMIP+)"

_The Cryosphere, 2019_

## Referee Comment (RC1) · Ralf Greve (Referee) · 18 Feb 2020

In this manuscript, Cornford et al. report the results from MISMIP+, the third in a series of intercomparison exercises for marine ice sheets. It features a 3D channel geometry that allows buttressing of grounded ice by an ice shelf and a retrograde bed slope in a part of the domain. Several experiments that produce grounding line retreat and re-advance are considered. The study benefits from a variety of contributing models/model variants that cover different treatments of the stress balance and basal sliding as well as different numerical techniques and resolutions. Overall, I found the results very interesting and the presentation adequate. I'd only like to raise some minor issues that should be dealt with as follows:

[Figure]

All multi-panel figures: I'd suggest to label the panels by a, b, etc., and refer to these labels in the captions, rather than using 'top left' etc.

Line 12, "All ice sheet models are based upon some approximation to Stokes flow": This might be misunderstood as no models use _full_ Stokes.

Lines 22-24, "community exercises comparing ice stream and shelf models": Seroussi et al. (2019) and Goelzer et al. (2018) are the InitMIP-Antarctica and Greenland papers, respectively. These exercises were about comparing _ice sheet_ models, which usually include ice stream and shelf dynamics, but the focus was not on the ice stream and shelf components.

Line 47: configuration_s_

Caption of Fig. 1: "MIMSIP+" -> "MISMIP+"

Line 50: Delete comma before "saw".

Line 59: "about" -> "with respect to" (?)

Line 105: "the Weertman (1957)" -> "the Weertman (1957) _rule/sliding law_"

Lines 105-123: Somewhere around here the findings of the study by Gladstone et al. (2017, https://doi.org/10.5194/tc-11-319-2017) should also be mentioned: keeping the basal friction continuous across the grounding line is beneficial for simulating marine ice sheets/grounding line dynamics. Weertman-Budd-type sliding (their Eq. (1)) is another alternative to achieve this.

Line 254: Missing space after "years".

Lines 273-274: The description of the Úa model could be a bit more comprehensive (along the lines of the other descriptions).

Figure 3, right panel: If I interpret this correctly, some of the results fall outside the required interval of $450 \pm 10$ km. Does this have any consequences?

Line 294: "lines intersect with"

Line 325, "450 ± 15 km": In section 2.2, a requirement of 450 ± 10 km was formulated. How does this go together?

Lines 327-329: What are "distinctive" vs. "conventional" numerical methods/treatments?

Lines 426-433: I'm not sure whether I got this right. Is the recommendation that sub-grid melt schemes should not be used?

Lines 439-440: "the distinction ... even between approximate models and full Stokes models, was minor": Fig. 9 (left panel) showed that there was a distinction ("much lower rate of advance"). This seems to be contradictive.

---

## Referee Comment (RC2) · Frank Pattyn (Referee) · 19 Feb 2020

**Overall assessment**

This paper reports on a follow-up of a series of marine ice sheet model intercomparisons (MISMIP) in which particular attention is paid to the effect of buttressing on the stability of grounding lines on retrograde slopes. The geometry of the experiment is taken from Gudmundsson et al (2013), i.e., a narrow overdeepened channel in the marine portion of the grounded ice sheet. Several models, from full Stokes to models of intermediate complexity (including membrane stresses in a variety of ways) participated in the experiment that consist of applying sub-shelf melt starting from an initial steady-state configuration on a retrograde bedrock slope. All participating models show

the same qualitative behaviour, meaning that a steady-state configuration across a retrograde section of the bed is obtained due to buttressing and grounding line retreat is initiated when sub-shelf melt is applied. The paper is definitely interesting for publication in TC and will become a benchmark for future model development. Nevertheless, I have a few points that need some clarification.

From the start (abstract) the authors promote the experiment as a test for the treatment of viscous stress sufficient for buttressing. However, throughout the paper, the emphasis is on the response of the different models in relation to basal sliding rules and differences in stress balance, but further discussion on buttressing remains absent. What are the implications of the results in our understanding of buttressing of ice flow on retrograde slopes?

It is also important to note that all models qualitatively show the same results. It is not necessary to dive immediately into their differences. It is important to highlight that models are capable of reproducing stable grounding lines on retrograde bed slopes in case of strong buttressing, irrespective of sliding rules or approximation to the stress balance.

Another point is that the majority of models are from the intermediate complexity group, i.e. higher-order models that include membrane stresses one way or the other. The authors particularly focus on two outliers, a HySSA model and two full Stokes models (Elmer/Ice and ISSM). Most of the attention goes to the HySSA model, but the similar behaviour of the two FS models is intriguing but left out without further discussion, especially since these are the models containing most of the physics. Both FS models are different and use different sliding laws/rules. One of them also solves for L1Lx and has a comparable behaviour to the other L1Lx models, showing that numerical issues should not be the culprit. The minor advance in the Ice1ra experiment therefore requires more discussion in the context of why the other (L1Lx) models advance so much more in this particular experiment when sub-shelf melt is halted. It could inform more on viscous stress important in buttressing, for instance.

**Detailed remarks**

Line 3: what is meant by 'sufficient' buttressing? Please also state what is meant by the testing of the models. What is tested and what for? It is not clear what the precise goal of the experiments is, especially in relation to buttressing (see major remark).

Line 40: strongly buttresses instead of buttresses

Line 47: configurations

Line 60: This was also the case for the MISMIP3d experiments

Line 104: suggest to continue the sentence: where $u$ is the horizontal ...

Line 105: Weertman (1957) sliding law/rule

Line 107: is continuous

Line 109, eq 4: define $\alpha^2$

Line 130: this is not the definition of steady-state. can it be mentioned over what time scales the models were run to reach the 'pseudo steady state'?

Line 188: Has this controversial modification been applied by any of the participating models? In either case, it should be stated.

Line 244: 'the use or otherwise': I don't understand this sentence. Moreover, it is not clear where and how this transformation is applied and Bueler et al (2015) does not shed further light on this. Some more details should be given, as this is also mentioned further down in the manuscript.

Line 274: in comparison with the other models, this description is really short. For the sake of balance, some more details should be given.

Line 286-289: Any reason why this is the case? This should be mentioned.

Line 297: Last sentence: does this apply to the Ice0 experiment?

[Figure]

Line 349-350: See major remark. It is not because they are underrepresented that you are loath to make much of these differences. I would say that L1Lx models are over-represented in this case.

Line 355-356: Or there are too many models of a particular category.

Line 388: This doesn't seem to apply to the FS models it seems.

Line 396-397: See major remark: It is important to note that all models qualitatively show the same results and to highlight that models are capable of reproducing stable grounding lines on retrograde bed slopes in case of strong buttressing, irrespective of sliding rules or approximation to the stress balance. Even the HySSA qualitatively shows a similar behaviour; its exception is can then be further discussed as is done in this paragraph.

Line 422-425: There is also no Coulomb-limiting factor in MISMIP3d. Furthermore, the friction coefficient in MISMIP3d leads to a sharp contrast between grounded and floating ice (higher friction that in this experiment).

Discussion: more discussion on the FS models is needed as well as buttressing in general (see major remark).

Line 439-440: I wouldn't say that the distinction between FS and other models is minor. The Ice1ra experiment proves otherwise.

Line 442: Either add that HySSA has qualitatively the same behaviour, or more general state in the summary that all models exhibit qualitatively the same behaviour and are capable of reproducing stable grounding lines on retrograde slopes in case of strong buttressing.

---

## Referee Comment (RC3) · Fuyuki SAITO (Referee) · 21 Feb 2020

This paper present the results of MISMIP+, a new model intercomparision of marine ice sheet models. I think this paper is fairly well written with some exception below, and can be accepted with minor revision.

L23 to L38: MISMIP+ is the first experiment among MISMIP series, which changes the mass balance. I think this should be emphasized.

Figure 1. Remark at top or bottom as t=0 may help. The vertical directions show a schematic view of ice volume (bigger ice sheet towards upper), which can be mentioned. The subexperiment names (Ice1r, Ice1rr, Ice1ra etc) should be mentioned in the figure.

[Figure]

Eq (1) and Table 1. I try to draw the bedrock map using the equation and values in the table, but I cannot reproduce the figure. I suspect the signs of Bn are all opposite. If that is correct, evelation around (0,-40) is higher than +300m, which is not clear using the present contours.

L46 in Asay-Davis et al. (2016).

L56. Need to include somewhere around here that the x-direction is toward the flow while the y is across, or at least refer Figure 2.

Eqs.(4) (5): alpha is not defined.

Eq.(7) confusing. max is outside tanh, but not clear.

L145. It is not clear whether m2 is applied only at the base of ice shelf or not. It matters for the regions close to the lateral walls.

Figure 3. Horizntal labels are missing. x(km) and t(yr)?

Figures 4 and 11 top left. Ice1r (t=0) and Ice2r (t=0) is identical by definition, which may be better to mention.

L313 less than 10km. But, this value is correct? It looks around 20km spread for yellow lines.

SAITO Fuyuki.

---

## Author Comment (AC1) · 13 Apr 2020

**Response to Prof. Greve's review**

We thank Prof. Greve for this supportive and thorough review, and acknowledge all of the points made. We reply to each point made (black text) below using blue text.

In this manuscript, Cornford et al. report the results from MISMIP+, the third in a series of intercomparison exercises for marine ice sheets. It features a 3D channel geometry that allows buttressing of grounded ice by an ice shelf and a retrograde bed slope in a part of the domain. Several experiments that produce grounding line

[Figure]

retreat and re-advance are considered. The study benefits from a variety of contributing models/model variants that cover different treatments of the stress balance and basal sliding as well as different numerical techniques and resolutions. Overall, I found the results very interesting and the presentation adequate. I'd only like to raise some minor issues that should be dealt with as follows:

All multi-panel figures: I'd suggest to label the panels by a, b, etc., and refer to these labels in the captions, rather than using 'top left' etc.

Agreed and done

Line 12, "All ice sheet models are based upon some approximation to Stokes flow": This might be misunderstood as no models use _full_ Stokes.

Rephrased to read "All ice sheet models are based upon Stokes flow or, more commonly, one of several approximations to Stokes flow"

Lines 22-24, "community exercises comparing ice stream and shelf models": Seroussi et al. (2019) and Goelzer et al. (2018) are the InitMIP-Antarctica and Greenland papers, respectively. These exercises were about comparing _ice sheet_ models, which usually include ice stream and shelf dynamics, but the focus was not on the ice stream and shelf components.

Rephrased: "community exercises comparing ice sheet models where ice stream and ice shelf dynamics are important.."

Line 47: configuration_s_

corrected

Caption of Fig. 1: "MIMSIP+" -> "MISMIP+"

corrected

Line 50: Delete comma before "saw".

corrected

Line 59: "about" -> "with respect to" (?)

'Àbout' seems correct to us.

Line 105: "the Weertman (1957)" -> "the Weertman (1957) rule/sliding law"

added "sliding law"

Lines 105-123: Somewhere around here the findings of the study by Gladstone et al. (2017, https://doi.org/10.5194/tc-11-319-2017) should also be mentioned: keeping the basal friction continuous across the grounding line is beneficial for simulating marine ice sheets/grounding line dynamics. Weertman-Budd-type sliding (their Eq. (1)) is another alternative to achieve this.

Added a footnote to acknowledge this paper while keeping the flow of the paragraph

Line 254: Missing space after "years".

corrected

Lines 273-274: The description of the Úa model could be a bit more comprehensive (along the lines of the other descriptions).

A more comprehensive description was included

Figure 3, right panel: If I interpret this correctly, some of the results fall outside the required interval of $450 \pm 10$ km. Does this have any consequences?

Yes - this is why the Weertman models have a wider spread of retreat rates. This is discussed in the paragraph beginning 'One major division'

Line 294: "lines intersect with"

corrected

Line 325, "$450 \pm 15$ km": In section 2.2, a requirement of $450 \pm 10$ km was formulated.

[Figure]

How does this go together?

Added a footnote: The requirement that $x = 450 \pm 10$ km was relaxed since many participants neglected it.

Lines 327-329: What are "distinctive" vs. "conventional" numerical methods/treatments?

We removed these adjectives

Lines 426-433: I'm not sure whether I got this right. Is the recommendation that subgrid melt schemes should not be used?

We do indeed think that such schemes should be avoided, or at least treated with great care.

Lines 439-440: "the distinction ... even between approximate models and full Stokes models, was minor": Fig. 9 (left panel) showed that there was a distinction ("much lower rate of advance"). This seems to be contradictive.

This was rephrased and a discussion of the full Stokes model has been added. The picture is not quite as simple as saying the Stokes models are different and better (though that is of course one possibility). Notably, although there are point of disagreement between the Stokes and non-Stokes models, there are also differences of similar magnitude between the two Stokes models (Elmer/Ice FS and ISSM FS). Elmer/Ice FS shows essentially the same rate of retreat in Ice1r as the SSA/HO/L1Lx models, while ISSM FS shows the same average rate but a much greater initial rate and lower final rate. Elmer/Ice FS shows no re-advance at all, which seems likely to be a numerical issue, while ISSM FS shows re-advance that is slower than average but comparable to its ISSM SSA counterpart.

Please also note the supplement to this comment:
https://www.the-cryosphere-discuss.net/tc-2019-326/tc-2019-326-AC1-

supplement.pdf

---

## Author Comment (AC2) · 13 Apr 2020

**Response to Prof. Saito's review**

We thank Prof Saito for this supportive and thorough review and have made a number of modifications in response. We respond to specific points made in the review (black text) in blue text below.

This paper present the results of MISMIP+, a new model intercomparision of marine ice sheet models. I think this paper is fairly well written with some exception below, and can be accepted with minor revision.

L23 to L38: MISMIP+ is the first experiment among MISMIP series, which changes the

[Figure]

mass balance. I think this should be emphasized.

We added this point to the paragraph beginning in L39

Figure 1. Remark at top or bottom as t=0 may help. The vertical directions show a schematic view of ice volume (bigger ice sheet towards upper), which can be mentioned. The subexperiment names (Ice1r, Ice1rr, Ice1ra etc) should be mentioned in the figure.

t =0, t = 100, t = 200 markers were added to the diagram. The sub-experiment names were added too. The relationship with vertical direction and grounding line advance/retreat was mentioned in the caption - it seemed misleading to place e.g an up arrow on the diagram because it might suggest that the experiments start with different ice volumes.

Eq (1) and Table 1. I try to draw the bedrock map using the equation and values in the table, but I cannot reproduce the figure. I suspect the signs of Bn are all opposite. If that is correct, evelation around (0,-40) is higher than +300m, which is not clear using the present contours.

Yes, the signs were confused. More contours (+5 m, and + 205 m) were added so that it is clear that elevation around (0,-40) is well above sea level (+349 m is the maximum value)

L46 in Asay-Davis et al. (2016).

corrected

L56. Need to include somewhere around here that the x-direction is toward the flow while the y is across, or at least refer Figure 2.

Added a sentence: Ice flows in a direction roughly parallel to the $x-$axis from $x = 0$ toward x=640 km, so that we will refer to $x$ as the along flow direction and $y$ as the lateral or across flow direction.
Eqs.(4) (5): alpha is not defined.

We added detail on $\alpha^2$, the coefficient of friction in a Coulomb friction law.

Eq.(7) confusing. max is outside tanh, but not clear.

A bracket was introduced to more clearly delimit the argument of $\tanh$: $m_1 = 0.2 \tanh\left(\frac{z_d - z_b}{75}\right) \max(-100 - z_d, 0)$

L145. It is not clear whether m2 is applied only at the base of ice shelf or not. It matters for the regions close to the lateral walls.

A clarification was added to the definition of $m_2$, stating that is was applied to ice shelf regions only.

Figure 3. Horizntal labels are missing. x(km) and t(yr)?

Yes - corrected

Figures 4 and 11 top left. Ice1r (t=0) and Ice2r (t=0) is identical by definition, which may be better to mention.

A note was added to the caption of fig 11, which now includes a sentence: The Ice2r experiment starts (as does the Ice1r experiment, see fig. 4) from the Ice0 steady state position

L313 less than 10km. But, this value is correct? It looks around 20km spread for yellow lines.

Yes, around 20 km. The text was modified accordingly
* * *

---

## Author Comment (AC3) · 13 Apr 2020

**Response to Prof. Pattyn's review**

We thank Prof. Pattyn for this supportive and thorough review, and acknowledge all of the points made. We reply to each point made (black text) below using blue text.

hdr

**Overall assessment**

This paper reports on a follow-up of a series of marine ice sheet model intercomparisons (MISMIP) in which particular attention is paid to the effect of buttressing on the stability of grounding lines on retrograde slopes. The geometry of the experiment is taken from Gudmundsson et al (2013), i.e., a narrow overdeepened channel in the marine portion of the grounded ice sheet. Several models, from full Stokes to models of intermediate complexity (including membrane stresses in a variety of ways) participated in the experiment that consist of applying sub-shelf melt starting from an initial steady-state configuration on a retrograde bedrock slope. All participating models show the same qualitative behaviour, meaning that a steady-state configuration across a retrograde section of the bed is obtained due to buttressing and grounding line retreat is initiated when sub-shelf melt is applied. The paper is definitely interesting for publication in TC and will become a benchmark for future model development.

Nevertheless, I have a few points that need some clarification. From the start (abstract) the authors promote the experiment as a test for the treatment of viscous stress sufficient for buttressing. However, throughout the paper, the emphasis is on the response of the different models in relation to basal sliding rules and differences in stress balance, but further discussion on buttressing remains absent. What are the implications of the results in our understanding of buttressing of ice flow on retrograde slopes?

This is a good point, but we would suggest that the common points and differences between the models do not reveal much about buttressing per se, beyond results that are already known: buttressing can lead to steady equilibrium with the grounding line crossing a partially retrograde bed, and the loss of that buttressing through ice shelf thinning can lead to grounding line retreat. We have rephrased and expanded the abstract (below, with deletions and additions highlighted) to make this clear

'We present the result of the third Marine Ice Sheet Intercomparison project, MISMIP+. MISMIP+ is intended to be a benchmark for ice flow models which include fast sliding

marine ice streams and floating ice shelves and in particular a treatment of viscous stress that is sufficient  to model *buttressing*, where upstream ice flow is restrained by a downstream ice shelf.  A set of idealized experiments  first test that models are able to maintain a steady state with the grounding line located on a retrograde slope due to buttressing and then explore scenarios where a reduction in that buttressing causes ice stream acceleration, thinning, and grounding line retreat. The majority of participating models passed the first test and then produced similar responses to the loss of buttressing.  We find that the most important distinction between models in this particular type of simulation is in the treatment of sliding at the bed, with other distinctions – notably the difference between the simpler and more complete treatments of englacial stress, but also the differences between numerical methods – taking a secondary role.'

It is also important to note that all models qualitatively show the same results.  It is not necessary to dive immediately into their differences. It is important to highlight that models are capable of reproducing stable grounding lines on retrograde bed slopes in case of strong buttressing, irrespective of sliding rules or approximation to the stress balance.

Agreed - we now state this in the abstract (see previous response), and also begin the discussion with a the details of this important point

Another point is that the majority of models are from the intermediate complexity group,i.e. higher-order models that include membrane stresses one way or the other. Theauthors particularly focus on two outliers, a HySSA model and two full Stokes models(Elmer/Ice and ISSM). Most of the attention goes to the HySSA model, but the similar behaviour of the two FS models is intriguing but left out without further discussion, especially since these are the models containing most of the physics. Both FS models are different and use different sliding laws/rules.

We have added text to both the results and discussions sections, and a figure fig.1

in this response) considering the differences within the Stokes models and with their nearest non-full-Stokes models (those using the same code and sliding law). The picture is not quite as simple as saying the Stokes models are different and better (though that is of course one possibility). Notably, although there are point of disagreement between the Stokes and non-Stokes models, there are also differences of similar magnitude between the two Stokes models (Elmer/Ice FS and ISSM FS). Elmer/Ice FS shows essentially the same rate of retreat in Ice1r as the SSA/HO/L1Lx models, while ISSM FS shows the same average rate but a much greater initial rate and lower final rate. Elmer/Ice FS shows no re-advance at all, which seems likely to be a numerical issue, while ISSM FS shows re-advance that is slower than many models but similar to ISSM SSA.

One of them also solves for L1Lx and has a comparable behaviour to the other L1Lx models, showing that numerical issues should not be the culprit.

We suggest (but of course do not know) that numerical issues could still be the culprit. The full Stokes and SSA/L1Lx/HO problems are different, with the Stokes problem being more difficult, even if other aspects of the numerical treatment are the same. Stokes models must solve for the scalar pressure to satisfy $\vec{\nabla}.\vec{u} = 0$ (while lesser models have it easy, so to speak) and also solve a more complicated contact problem. See e.g Gagliardini 2015 for a case where a numerical decision affects Elmer/Ice quite strongly at even 50 m resolution in MISMIP3d

The minor advance in the Ice1ra experiment therefore requires more discussion in the context of why the other (L1Lx) models advance so much more in this particular experiment when sub-shelf melt is halted. It could inform more on viscous stress important in buttressing, for instance.

A discussion has been added (see above)

**Detailed remarks**

Line 3: what is meant by 'sufficient' buttressing? Please also state what is meant by the testing of the models. What is tested and what for? It is not clear what the precise goal of the experiments is, especially in relation to buttressing (see major remark).

We meant that the models were sufficient, rather than the buttressing. To clarify this, we changed the phrase to "a treatment of viscous stress that is sufficient **to model** buttressing. We also clarified the remainder of the abstract to say what is tested (see response to major remark) and what is explored, which we think then makes the goal clearer.

Line 40: strongly buttresses instead of buttresses

Agreed, and done

Line 47: configurations

corrected

Line 60: This was also the case for the MISMIP3d experiments

Noted, with citation

Line 104: suggest to continue the sentence: where u is the horizontal ...

Done

Line 105: Weertman (1957) sliding law/rule

added "sliding law"

Line 107: is continuous

Yes

Line 109, eq 4: define $\alpha$

We added detail on $\alpha^2$, the coefficient of friction in a Coulomb friction law.

Line 130: this is not the definition of steady-state. can it be mentioned over what time scales the models were run to reach the 'pseudo steady state'?

Added some wording to make this point clear, and also added 'Participants were free to produce the initial state by any method; the majority of participants chose to evolve their models with the stated parameters for $\sim 10$ ka in order to approach steady state'

Line 188: Has this controversial modification been applied by any of the participating models? In either case, it should be stated.

Yes, though the models in question have in general avoided the modification since. We added 'Several participants employed such a modification in these exercises, with clear consequences in the Ice2 experiments (Sec. 4.3)'

Line 244: 'the use or otherwise': I don't understand this sentence. Moreover, it is not clear where and how this transformation is applied and Bueler et al (2015) does not shed further light on this. Some more details should be given, as this is also mentioned further down in the manuscript.

This is described in the cited text (Beuler et al 2005) in section 5.2 (Regularization of the margin by a transformation). We noted the section in the text.

Line 274: in comparison with the other models, this description is really short. For the sake of balance, some more details should be given.

A more comprehensive description was included.

Line 286-289: Any reason why this is the case? This should be mentioned.

There is more explanation in the individual sections - this paragraph is an overview before the detailed results

Line 297: Last sentence: does this apply to the Ice0 experiment?

The sentence 'Not every submission included this test, although the majority did?'. Yes - one or two submissions omitted Ice0, notably the Elmer/Ice full Stokes model. I (SLC) judged that the Elmer/Ice full Stokes contribution was too important to exclude on these grounds.

Line 349-350: See major remark. It is not because they are underrepresented that you are loath to make much of these differences. I would say that L1Lx models are over-represented in this case.

A fair point - see the response to the major remark, but in brief we have added some more discussion of the full Stokes models

Line 355-356: Or there are too many models of a particular category.

Too many L1Lx/SSA models? We agree that this class of model is common

Line 388: This doesn't seem to apply to the FS models it seems.

A fair point - see the response to the major remark, but in brief we have added some more discussion of the full Stokes models

Line 396-397: See major remark: It is important to note that all models qualitatively show the same results and to highlight that models are capable of reproducing stable grounding lines on retrograde bed slopes in case of strong buttressing, irrespective of sliding rules or approximation to the stress balance. Even the HySSA qualitatively shows a similar behaviour; its exception is can then be further discussed as is done in this paragraph.

Agreed - the discussion now begins with a paragraph emphasising that point.

Line 422-425: There is also no Coulomb-limiting factor in MISMIP3d. Furthermore, the friction coefficient in MISMIP3d leads to a sharp contrast between grounded and floating ice (higher friction that in this experiment).

We added the details on friction. There are as many MISMIP+ models that do not

employ the Coulomb limit as do, so that in itself does not explain the difference.

Discussion: more discussion on the FS models is needed as well as buttressing in general (see major remark).

A fair point - see the response to the major remark, but in brief we have added some more discussion of the full Stokes models

Line 439-440: I wouldn't say that the distinction between FS and other models is minor.The Ice1ra experiment proves otherwise.

A fair point - see the response to the major remark, but in brief we have added some more discussion of the full Stokes models

Line 442: Either add that HySSA has qualitatively the same behaviour, or more general state in the summary that all models exhibit qualitatively the same behaviour and are capable of reproducing stable grounding lines on retrograde slopes in case of strong buttressing.

Agreed - we added this note, but also began the discussion (as suggested above) by noting the important point that models were more similar than different (with the possible exception of the Stokes models

[Figure]

**Fig. 1.** Comparison of full Stokes and vertically integrated models in the Ice1r and Ice1ra experiments. Panel (a) shows the change in grounded area, panel (b) the change in volume above flotation.